# Let's Grow an Unbiased Community : Guiding the Fairness of Graphs via New Links

## Abstract

Graph Neural Networks (GNNs) have achieved remarkable success across diverse applications. However, due to the biases in the graph structures, graph neural networks face significant challenges in fairness. Although the original user graph structure is generally biased, it is promising to guide these existing structures toward unbiased ones by introducing new links. The fairness guidance via new links could foster unbiased communities, thereby enhancing fairness in downstream applications. To address this issue, we propose a novel framework named FairGuide. Specifically, to ensure fairness in downstream tasks trained on fairness-guided graphs, we introduce a differentiable community detection task as a pseudo downstream task. Our theoretical analysis further demonstrates that optimizing fairness within this pseudo task effectively enhances structural fairness, promoting fairness generalization across diverse downstream applications. Moreover, FairGuide employs an effective strategy which leverages meta-gradients derived from the fairness-guidance objective to identify new links that significantly enhance structural fairness. Extensive experimental results demonstrate the effectiveness and generalizability of our proposed method across a variety of graph-based fairness tasks. The code and datasets are available at
https://anonymous.4open.science/r/FairGuide-8907/README.md.

## 1 INTRODUCTION

Graph data has become an essential part of many applications like social networks analysis Kumar et al. (2022), recommendation systems Wu et al. (2022), and fraud detection Cheng et al. (2020). In graph data, nodes typically represent entities or individuals, while edges capture the relationships between them. Graph Neural Networks (GNNs) have emerged as powerful tools for leveraging both node features and graph topology to perform tasks such as node classification Kipf & Welling (2016); Rong et al. (2020), graph embedding Ying et al. (2018); Zhu et al. (2020), and link prediction Zhang & Chen (2018), leading to significant improvements in task performance.

Despite the great performance of GNNs, linking biases in user graphs raise fairness concerns when deploying GNNs in critical applications. Specifically, as illustrated in Fig. 1, online social networks and residential communities often exhibit pronounced structural barriers, with tightly interconnected subgroups interacting primarily within themselves, resulting in inequitable access to resources, opportunities,and influence Saxena et al. (2024). Such biases originating from data can be further amplified by

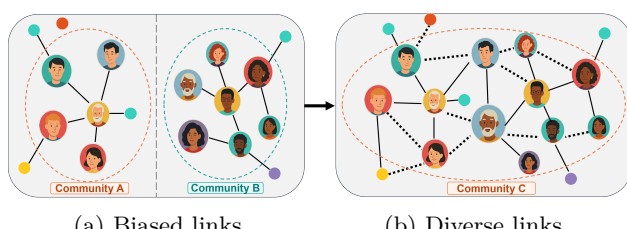

(a) Biased links      (b) Diverse links

Figure 1: Guiding fairness via new links.

the message-passing mechanisms of GNNs. For example, social network recommendation systems have been found to systematically prevent female profiles from becoming among the most commented on or liked Bose & Hamilton (2019). Similarly, GNN-based book recommendation systems can be biased toward recommending books authored by males, thereby reinforcing existing disparities and limiting exposure to diverse content Buyl & De Bie (2020).

Extensive research has been conducted to address fairness issues in graph neural networks. Specifically, in-processing Buyl & De Bie (2020); Wang et al. (2022); Zhao et al. (2022); Zhu et al. (2024a;b); Ling et al. (2023); Luo et al. (2024) and post-processing Kang et al. (2020); Masrour et al. (2020); Bose & Hamilton (2019) methods modify the training procedure or final predictions of a particular GNN model, and thus can improve fairness for a specified task. However, they do not remove the biased graph itself. Once the same graph is reused by another GNN or another task, the bias may still be propagated by graph structure.

Pre-processing methods operate on graph data prior to model training, but existing methods are often designed to synthesize a fair training set rather than providing feasible modifications for existing graphs. For instance, methods like EDITS, FairGen, and FairDrop Dong et al. (2022); Zheng et al. (2024); Spinelli et al. (2021) rely on substantial graph reconstruction, node attribute modification, or edge deletion. These operations may be effective for constructing fair training inputs but are impractical for realworld user-graph, as they alter user profiles and discard valuable interactions. Furthermore, fair representation methods like Graphair Ling et al. (2023) only learn debiased embeddings, which are incompatible with structure-based downstream tasks like community detection.

Therefore, achieving graph fairness requires a practical fair guidance problem: *Can we guide existing real-world user graphs toward unbiased structures, thereby ensuring fairness in the deployment of GNN-based applications on these graphs?*. Link addition provides a natural mechanism for this, enabling a guidance framework to recommend new links while preserving existing relations and attributes. However, generating such guidance is non-trivial and presents two primary challenges:(i) **Task-agnostic guidance.** The fairness guidance aims to reduce the inherent structural biases by selectively suggesting new links. Therefore, guidance on graph growth toward fairness should be task-agnostic. It remains challenging to effectively guide user graphs toward fairness without knowledge of specific downstream tasks. (ii) **Strict intervention budget.** As discussed, users are unlikely to accept a large number of link suggestions. Thus, another crucial challenge is how to effectively guide graphs toward fairness by introducing only a limited number of new links.

To address these challenges, we propose a novel framework named FairGuide. Specifically, to eliminate structural biases without specific downstream tasks, FairGuide leverages community detection as a pseudo downstream task. Since labels in many downstream tasks are closely correlated with community structures, ensuring structural fairness for community detection intuitively benefits various downstream applications. This intuition is further supported by our theoretical analysis. To effectively utilize the limited link-addition budget, FairGuide employs a module to identify optimal new links for structural fairness. Specifically, an efficient meta-gradient computation method is deployed to approximate the impact of potential link additions on structural fairness. This enables recommending optimal links to guide the graph toward fairness. In summary, our main contributions are:

- We focus on a novel problem of fairness guidance, which aims to guide the existing biased graph structures into fair community via introducing new links.
- We propose a novel fairness guidance framework named FairGuide, which incorporates a pseudo downstream task and link addition through meta gradients to identify optimal new links for structural fairness.
- We collect a new large-scale social network from GitHub to provide empirical validation of FairGuide in a real-world scenario.
- Both theoretical analysis and empirical results demonstrate that our FairGuide can effectively guide graphs toward fairness to facilitate the fairness of downstream tasks.

## 2 RELATED WORK

In this section, we provide a comprehensive review of the literature on fair graph learning and link prediction, which are closely related to our work.

### 2.1 Fair Graph Learning

Graph learning models have been widely adopted for analyzing topological data, demonstrating outstanding performance in various graph-based tasks (Kipf & Welling, 2016; Zhang & Chen, 2018; Tsitsulin et al., 2023). However, recent studies (Kang et al., 2020; Ma et al., 2022) reveal that fairness concerns emerge

prominently in graph learning models. For instance, FairGNN (Dai & Wang, 2021) demonstrates that biases can implicitly propagate through graph structures, while EDITS (Dong et al., 2022) further demonstrates that biased graph topologies directly lead to discriminatory model outcomes. To address such challenges, numerous approaches have been proposed to improve fairness in graph learning. In-processing methods typically employ techniques like adversarial training (Ling et al., 2023; Chen et al., 2025), fairness-aware regularization (Bose & Hamilton, 2019; Jiang et al., 2024), invariant learning (Zhu et al., 2024a) to reduce bias of specific models. Some works also try to learn a fair graph representation by forcing distribution alignment (Li et al., 2024a) or preventing sensitive information leaking (Zhu et al., 2024b). Additionally, pre-processing strategies modify the graph data itself, such as edge rewiring (Spinelli et al., 2021; Li et al., 2021; Wang et al., 2025), feature masking (Ling et al., 2023) and graph reconstruction (Dong et al., 2022) to reduce the data distribution gap of different group and create fair input graphs for downstream tasks. Some data-oriented methods also achieve fairness at a lower cost through data rebalancing (Li et al., 2024b).

However, existing approaches face inherent limitations in solving the problem of guiding the fairness of graphs. In-processing methods are limited to debiasing specific graph neural networks and downstream tasks, making them inapplicable for addressing biases in the graph structures underlying graph data. Pre-processing methods attempt to achieve fairness by modifying graph data directly but often involve impractical link removal and lack explicit constraints on the amount of structural modifications, which is not suitable for the real-world community. By contrast, our method explores to obtain an unbiased graph network by guiding the original graph data towards a fairer state, which remains under-explored for prior works.

## 2.2 Link Prediction

Standing as one of the most fundamental tasks in graph representation learning, link prediction has been widely used in social network to help users discover and connect with individuals they have not yet interacted with or encountered (Su et al., 2020; Daud et al., 2020). Existing approaches to link prediction can be broadly categorized based on their underlying techniques, including similarity-based methods (Yu et al., 2017; Rossi et al., 2021), dimensionality reduction-based methods (Du et al., 2020) and other emerging paradigms such as graph neural networks and probabilistic models. Among these, similarity-based methods are the most prevalent and can be further divided into two paradigms: structure-based methods (Aziz et al., 2020; Luo et al., 2021) and attribute-aware methods (Xiao et al., 2021; Zhang et al., 2023). Structure-based methods rely on topological similarity metrics such as common neighbors to infer potential links while attribute-aware methods incorporate node features or semantic attributes to enhance prediction accuracy and capture more nuanced relationships. However, such similarity-based approaches face inherent fairness-related limitations. Due to the inherent similarity in the same group, traditional link prediction techniques are difficult to break the bubbles in the community (Masrour et al., 2020) and may even enlarge the bias. Some works have investigated fairness-aware link prediction or recommendation systems to mitigate algorithmic bias and improve diverse in recommended connections (Li et al., 2021; 2022). However, these works primarily focus on addressing specific link prediction or recommendation tasks, rather than providing systematic frameworks to help guide a fair and diverse graph network.

# 3 PRELIMINARY ANALYSIS

In this section, we present preliminaries of the fairness issues in the real-world user graph structures.

## 3.1 Notations

We use $\mathcal{G} = (\mathcal{V}, \mathcal{E}, \mathbf{X})$ to denote user graph where $\mathcal{V} = \{v_1, ..., v_N\}$ is the set of $N$ nodes, representing the users in the network, $\mathcal{E} \in \mathcal{V} \times \mathcal{V}$ is the set of edges, and $\mathbf{X} \in \mathbb{R}^{N \times M}$ is the node attribute matrix, and each node $v_i$ is associated with a $M$-dimensional feature vector $\mathbf{x}_i$. $\mathbf{A} \in \mathbb{R}^{N \times N}$ is the adjacency matrix of $\mathcal{G}$, where $\mathbf{A}_{ij} = 1$ if nodes $v_i$ and $v_j$ are connected, otherwise $\mathbf{A}_{ij} = 0$. $Y$ and $\hat{Y}$ represent the ground truth and outcomes for the downstream task, respectively. In this work, we focus on binary sensitive attribute which is denoted as $s \in \{0, 1\}$.

Table 1: Discrimination of GCN in node classification and CD tasks on GitHub.

| Tasks | Metrics | Original | Link Pred. | Rand. Add |
|---|---|---|---|---|
| Node Classification | F1 (%) ↑ | 78.6 ± 0.2 | 78.8 ± 0.1 | 78.0 ± 0.1 |
| | AUC (%) ↑ | 85.4 ± 0.5 | 85.4 ± 0.1 | 84.3 ± 0.1 |
| | $\Delta_{SP}$(%) ↓ | 12.5 ± 0.4 | 11.9 ± 0.2 | 11.0 ± 0.2 |
| | $\Delta_{EO}$(%) ↓ | 8.5 ± 0.5 | 8.3 ± 0.4 | 8.3 ± 0.3 |
| CD | $\Delta_{SP}$(%) ↓ | 41.9 ± 1.5 | 38.2 ± 4.0 | 35.7 ± 2.8 |

## 3.2 Collection of GitHub Dataset

For the purpose of this study, we constructed a real-world dataset by crawling from the social platform **GitHub**, which is the most popular open source platform in the world. This GitHub dataset contains more than 30,000 developers' profiles and 270,000 links between users. The features of developers include gender, region , followers, development language and etc. The links represent the follower-followee relationship between two developers. Both features and links of the dataset are collected by GitHub REST API. We divide users into *developed* and *developing* according to the countries they belong to and treat it as sensitive attribute. Then we further categorize a user with more than 35 followers as popular developer and treat it as the predicting target for the classification task.

## 3.3 Preliminaries of Fairness

We focus on two widely used group fairness notions, i.e., statistical parity Dwork et al. (2012) and equal opportunity Hardt et al. (2016). And we consider a binary sensitive attribute, i.e., $s \in \{0, 1\}$.

**Definition 1** (Statistical Parity). *Statistical parity ensures that the predicted result $\hat{Y}$ is independent of the sensitive attribute $s$. Formally, for a binary classification task, i.e, $\hat{Y} \in \{0, 1\}$, the metric of statistical parity is computed by:*

$$\Delta_{SP} = |P(\hat{Y} = 1|s = 0) - P(\hat{Y} = 1|s = 1)|. \tag{1}$$

*For a multi-category task, i.e., $\hat{Y} \in \{0, 1, ..., C\}$ , $\Delta_{SP}$ extends to:*

$$\Delta_{SP} = \frac{1}{2} \sum_{C_k \in \mathcal{C}} \left| P(\hat{Y} = C_k \mid s = 0) - P(\hat{Y} = C_k \mid s = 1) \right| \tag{2}$$

*A lower value of $\Delta_{SP}$ indicates fairer predictions with respect to the sensitive attribute.*

**Definition 2** (Equal Opportunity). *Equal opportunity requires that the predicted result $\hat{Y}$ is conditionally independent of the sensitive attribute $s$ , given the true label $Y$. Considering a binary classification task, the metric of equal opportunity is formulated as:*

$$\Delta_{EO} = |P(\hat{Y}|s = 0, Y = 1) - P(\hat{Y}|s = 1, Y = 1)| \tag{3}$$

*Similar to $\Delta_{SP}$, lower $\Delta_{EO}$ implies fairer results of predictions.*

## 3.4 Discrimination Analysis on GitHub Dataset

To analyze the biases of the user graph structure, we adopt GCN Kipf & Welling (2016) to perform node classification task and Louvain algorithm Blondel et al. (2008) to perform community detection task. From Tab. 1, we can observe that both $\Delta_{SP}$ and $\Delta_{EO}$ exhibit high values in the node classification and community detection tasks, indicating significant biases within the original graph structure.

Since we aim to guide the fairness of graphs via new links, we conduct preliminary analysis to investigate how simple link addition strategies will affect the performance and fairness on downstream tasks.Specifically, two link addition strategies are considered.(i) **Link Pred.**: It adds edges by a link predictor Schlichtkrull et al. (2018) ; (ii) **Rand. Add**: It randomly adds edges to the GitHub graph; Both strategies are constrained to add 4% of existing links. The results on node classification and community detection are presented in Tab. 1, where we can observe that:

- Adding edges via link prediction improves task performance but has limited effectiveness in mitigating bias. This is because connecting similar nodes primarily reinforces existing community structures, thus preserving structural biases;

- Adding some random links is slightly effective in mitigating bias. This is because these random edges can lead to inter-group connections. However, the biases are still significant and the task performance degrades largely.

These observations indicate that existing link prediction methods are insufficient to address the fairness guidance problem.

### 3.5 Problem Definition

The analysis presented in Sec. 3.4 demonstrates the necessity of developing methods that guide graph structures toward fairness by introducing new links. With the notations in Sec. 3.1 and fairness definitions in Sec. 3.3, the fairness guidance via link addition can be formulated as:

**Problem 1** (Fairness Guidance via Link Addition)**.** *Given a user graph $\mathcal{G} = (\mathcal{V}, \mathcal{E}, \mathbf{X})$ with sensitive attributes $\mathcal{S}$, our objective is to strategically add $n$ links to obtain a fair graph $\mathcal{G}' = (\mathcal{V}, \mathcal{E}', \mathbf{X})$. The number of added links, denoted as $n$, is subject to a constraint $\Delta$. A GNN model $f : \mathcal{G}' \to \hat{y}$, trained on the modified graph $\mathcal{G}'$, is required to produce predictions $\hat{y}$ that satisfy the fairness criteria introduced in Sec. 3.3.*

## 4 METHODOLOGY

In this section, we present the details of FairGuide. FairGuide formulates fairness guidance via new links as an optimization problem, which selects new links that minimize the structural biases. Two major challenges remain to be solved: (i) how to measure the biases of structures without accessing to the downstream tasks; (ii) how to add links to effectively guide the fairness of the user graph structures within a constrained budget for link addition. As illustrated in Fig. 2, FairGuide leverages community detection as a pseudo task to

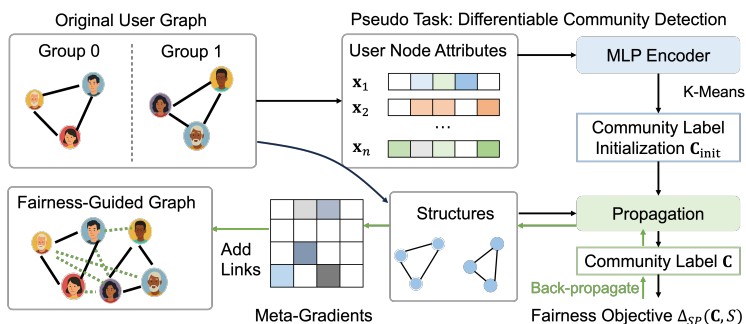

Figure 2: An illustration of FairGuide which adds new links to guide the growing of graph toward fairness.

reflect the structural biases for downstream tasks. This pseudo downstream task serves as a proxy for downstream tasks, allowing us to generalize fairness optimization without relying on task-specific adjustments. FairGuide computes meta-gradient scores to estimate the potential fairness effect of candidate links, and then applies a budget-constrained selection strategy to choose a limited number of high-scoring links. Next, we introduce the bi-level optimization objective function of FairGuide followed by details of each component.

### 4.1 Overall Goal of FairGuide

The goal of FairGuide is to add fairness-guiding links to $\mathcal{G}$ to ensure the GNN classifier trained on the updated graph structure $\mathcal{G}'$ produces fair predictions. Let $\mathcal{Y}_t$ and $\mathcal{L}_t$ denote the labels and the training loss for the target downstream task, respectively. Fairness guidance via the introduction of new links can be formulated as the following bi-level optimization on the structures:

$$
\begin{aligned}
\min_{\mathcal{G}'} \quad & \mathcal{M}\big(f_{\theta^*}(\mathcal{G}'), \mathcal{S}, \mathcal{Y}_t\big) \\
\text{s.t.} \quad & \theta^* = \arg\min_{\theta} \mathcal{L}_t\big(f_\theta(\mathcal{G}'), \mathcal{Y}_t\big), \quad \mathcal{E}' \supseteq \mathcal{E}, \quad \mathcal{E}' - \mathcal{E} \le \Delta
\end{aligned}
\tag{4}
$$

where $\mathcal{M}$ measures the fairness of $f_{\theta^*}$ on the updated graph structures $\mathcal{G}'$ that have been added links. $\mathcal{L}_t$ denotes the training loss of the downstream GNN classifier. The constraint $\mathcal{E}' - \mathcal{E} \le \Delta$ limits the number of

added links to at most $\Delta$. In the inner-level optimization, it simulates the training of the GNN classifier on the user graph for the target downstream task. In the outer-level optimization, the optimal set of new links are identified to promote fairness in the downstream GNN classifier.

## 4.2 Pseudo Downstream Task

With the Eq.(4), the fairness guidance via link addition is formatted to a bi-level optimization on the graph structures under constraints. However, the Eq.(4) requires to measure the fairness based on GNN predictions on downstream tasks, which are unavailable during the fairness guidance process. To address this problem, FairGuide deploys a pseudo task to reflect the fairness on the downstream task. Specifically, user communities naturally capture structural and node attribute information, and community labels often correlate strongly with labels from various downstream tasks. Thus, discrimination observed in an unsupervised community detection task can reflect structural biases relevant to downstream tasks. This intuition is further verified by the theoretical analysis. Therefore, community detection is deployed as the pseudo downstream task in FairGuide to estimate structural biases without the specific knowledge of downstream tasks. Next, we provide theoretical justification of adopting the pseudo downstream task for fairness on downstream tasks. Then, we present the updated objective function, followed by the design of community detection as the pseudo task.

**Theoretical Justification**. In the following, we present a theorem that justifies our motivation that alleviating the bias of downstream tasks can be achieved by reducing the correlation coefficient between the sensitive attribute $\mathcal{S}$ and the community label $\mathbf{C}$. Below, we first present the definition of the Pearson correlation coefficient followed by the theorem and proof.

**Definition 3.** *(Pearson Correlation Coefficient). Pearson correlation coefficient measures the linear correlation between two random variables $X$ and $Y$ as:*

$$\rho_{X,Y} = \frac{\mathbb{E}[(X - \mu_X) \cdot (Y - \mu_Y)]}{\sigma_X \cdot \sigma_Y} \tag{5}$$

**Theorem 1.** *Let $C$ and $S$ represent the community label and sensitive attribute, respectively. Let $\hat{Y}$ denote the output of a downstream prediction task. Assume that $C$ is highly correlated with $\hat{Y}$, i.e., $\rho_{C,\hat{Y}}$ is larger than a positive constant $\cos \alpha$. If the model is trained to make $\rho_{S,C}$ close to zero, i.e., within*

$$\left[ \cos \left( \frac{\pi}{2} + \delta \right), \cos \left( \frac{\pi}{2} - \delta \right) \right],$$

*where $\delta$ is close to 0, then the correlation between the sensitive attribute $S$ and the downstream prediction $\hat{Y}$ satisfies:*

$$\rho_{S,\hat{Y}} \in \left[ \cos \left( \frac{\pi}{2} + \delta + \alpha \right), \cos \left( \frac{\pi}{2} - \delta - \alpha \right) \right].$$

This theorem demonstrates that when bias mitigation in pseudo-community detection is achieved through link addition, the upper bound of bias in downstream tasks is consequently constrained. This theorem validates the deployment of community detection as the pseudo task to facilitate the structural fairness without the knowledge of downstream tasks.

We further discuss the connection between the correlation-based theorem and the statistical-parity objective in Appendix B.1.

**Updated Objective Function with Pseudo Downstream Task**. By adopting community detection as the pseudo task, the inner-level optimization is updated to the process of community detection. To measure the bias of the community labels $\mathbf{C}$, we adopt $\Delta_{SP}$ based on the predicted community assignments. Thus, the optimization problem in Eq.(4) can be reformulated as:

$$\min_{\mathcal{G}'} \Delta_{SP}(\mathbf{C}, \mathcal{S})$$
$$\text{s.t.} \quad \mathbf{C} = \text{CommunityDetection}(\mathcal{G}'),$$
$$\mathcal{E}' \supseteq \mathcal{E}, \quad \mathcal{E}' - \mathcal{E} \leq \Delta. \tag{6}$$

**Efficient Differentiable Community Detection**. Despite the advantages of using community detection as a pseudo task, traditional community detection methods are typically non-differentiable with respect to the graph structure. One may utilize GNN-based community detection $\mathbf{C} = f_{\theta'}(\mathcal{G}')$ with $\theta' = \arg\min_\theta \mathcal{L}_{CD}(f_\theta(\mathcal{G}'))$, where $\mathcal{L}_{CD}$ is the loss for community detection Sun et al. (2021). However, in this situation, computing the gradient with respect to the graph structure is computationally expensive, as it involves differentiating through the optimization process:

$$\frac{\partial \mathbf{C}}{\partial \mathcal{G}'} = \frac{\partial f_{\theta'}(\mathcal{G}')}{\partial \mathcal{G}'} + \frac{\partial f_{\theta'}(\mathcal{G}')}{\partial \theta'} \frac{\partial \theta'}{\partial \mathcal{G}'} \tag{7}$$

where the latter term $\frac{\partial \theta'}{\partial \mathcal{G}'}$ requires differentiating through the iterative training procedure of the GNN. To overcome this issue, we decouple the community detection into attribute-based clustering and structure-based aggregation. First, we utilize an MLP-based self-supervised auto-encoder to obtain latent feature representations for nodes. These representations are then clustered via K-means to generate initial community labels:

$$\mathbf{C}_{\text{init}} = \text{K-means}(\text{MLP}(\mathbf{X})). \tag{8}$$

The number of communities $C$ is the predefined hyperparameter. The MLP module operates as a self-supervised auto-encoder, optimized via a mean squared error (MSE) objective to reconstruct the original node features from the encoded latent representations.

Inspired by label propagation Zarezadeh et al. (2022); Gasteiger et al. (2019), we propose to aggregate the initial community labels by graph structure, enabling the incorporation of structural information. Specifically, the aggregation process for final community labels is described as:

$$\mathbf{C} = \text{softmax}\left((1-\alpha)^K \hat{\mathbf{A}}^K \mathbf{C}_{\text{init}} + \alpha \sum_{i=0}^{K-1} (1-\alpha)^i \hat{\mathbf{A}}^i \mathbf{C}_{\text{init}}\right), \tag{9}$$

where $\hat{A}$ is the symmetric normalized adjacency matrix and $\alpha$ is the probability of restarting from the original node features in aggregation. After $K$-layer aggregation, we get the final community labels. Since label propagation with structures is decoupled with the MLP-based community label initialization, the computation of $\frac{\partial \mathbf{C}}{\mathbf{A}}$ could be computed without differentiating through the MLP parameters, resulting in much more efficient gradient computation compared to Eq.(7).

### 4.3 Fairness-Guided Link Addition via Meta Gradients and Gumbel-max Sampling

In this subsection, we present how to optimize the updated objective function Eq.(6) given the differentiable community detection. Following previous works in updating structures Zügner & Günnemann (2019), we use meta-gradients as candidate-link scoring signals for the bi-level optimization problem in Eq.(6). Next, we first derive the computation of meta-gradients. Then, we present a strategy of link addition with meta-gradients that satisfies constraints on link discreteness and the number of new links.

**Meta-Gradients of Candidate Links**. The meta-gradient of a candidate link indicates how the addition of this link would influence the fairness of the pseudo community detection task. In the following, we formally illustrate the computation of the meta-gradient using the differentiable community detection method:

$$\nabla_{\mathbf{A}}^{\text{meta}} = \frac{\partial \Delta_{SP}(\mathbf{C}, \mathcal{S})}{\partial \mathbf{C}}\left(\frac{\partial \mathbf{C}}{\partial \mathbf{A}} + \frac{\partial \mathbf{C}}{\partial \mathbf{C}_{\text{init}}}\frac{\partial \mathbf{C}_{\text{init}}}{\partial \mathbf{A}}\right) \tag{10}$$

Here, the first term represents the direct influence of the perturbed graph structure $\hat{\mathbf{A}}$ on the final model output, while the second term accounts for the indirect effects propagated through initialized community labels. Note that initialized community labels are independent to the graph structure. Therefore, we can deduce that the indirect term $\frac{\partial \mathbf{C}_{\text{init}}}{\partial \mathbf{A}} = 0$. So finally the meta-gradient consequently be simplified as:

$$\nabla_{\mathbf{A}}^{\text{meta}} = \frac{\partial \Delta_{SP}(\mathbf{C}, \mathcal{S})}{\partial \mathbf{C}}\frac{\partial \mathbf{C}}{\partial \mathbf{A}}. \tag{11}$$

With the Eq.(11), the meta-gradients of candidate links can be efficiently computed. Next, we'll introduce how we optimize the adjacent matrix $\mathbf{A}$ discretely to satisfy the constraints of our optimization problem .

**Gumbel-max Sampling for Link Addition.** With the Eq.(11), we get meta-gradients of graph structures, i.e., $\nabla_{\mathbf{A}}^{\mathrm{meta}}$. And based on meta-gradient, a straightforward way to optimize the graph structure is:

$$\mathbf{A}' = \mathbf{A} - \alpha \nabla_{\mathbf{A}}^{\mathrm{meta}}. \tag{12}$$

However, this approach cannot guarantee the constraints of link discreteness and the maximum number of link addition. To overcome these issues, we reformulate edges adding process as a stochastic sampling process based on reweighted meta-gradients. According to the rule for the derivative of a scalar with respect to a matrix, the final gradient matrix $\nabla_{\mathbf{A}}^{\mathrm{meta}}$ has the same shape as the adjacency matrix $\mathbf{A}$ , and $\nabla_{\mathbf{A_{i,j}}}^{\mathrm{meta}}$ represents the possible effect of the edge i,j on fairness. Intuitively, to grow an unbiased community, more group-cross edges are needed to add, so we amplify gradients for edges connecting nodes with differing sensitive attributes while enabling probabilistic batch selection. Specifically, for each candidate edge $(i,j) \in \mathcal{E} = \{(u,v) \mid \mathbf{A}_{u,v} = 0\}$, we compute an adjusted gradient score:

$$\tilde{\nabla}i,j = -\nabla_{\mathbf{A_{i,j}}}^{\mathrm{meta}} \cdot (1 + \beta \cdot \mathbb{I}(s_i \neq s_j)), \tag{13}$$

Here $\beta$ controls inter-group connection incentives, and taking a negative value of the gradient means potential edges with smaller gradients have higher scores. We then sample edges via Gumbel-softmax reparameterization:

$$P(i,j) = \frac{\log(\tilde{\nabla}i,j + \epsilon) + g_{i,j}}{\tau}, \quad g_{i,j} \sim \mathrm{Gumbel}(0,1), \tag{14}$$

where $g_{i,j}$ is the gumbel noise, $\tau$ modulates exploration-exploitation trade-offs and $\epsilon$ prevents numerical underflow. And simultaneously the top-k edges with highest scores are added :

$$\mathbf{A}' = \mathbf{A} + \sum_{(i,j) \in \mathrm{Top}k(P(i,j))} \delta_{i,j}. \tag{15}$$

These formulas describe how meta-gradient scores are converted into discrete link additions under the edge budget. The meta-gradient computation estimates candidate-link effects, while the Gumbel-based top-$k$ selection enforces the discrete and budgeted update. After one round of the process of sampling edges to update the graph structure, the whole optimization pipeline goes back to aggregation step. Repeating the cycle for several times, we get the final structure $\mathbf{A}'$ with new fair links. The detailed algorithm of FairGuide can be found in Appendix A.

### 4.4 Time Complexity Analysis

In this section, we give the time complexity for adding a link for a specific node $v$. Specifically, the computational complexity of FairGuide is determined by two components, i.e., differentiable community detection, and fairness-guided link addition via meta gradients. To recommend a link for node $v$, the differentiable community detection involve the existing links and candidate links in the differential aggregation phase. The time complexity would be $O((d+1)c|\mathcal{V}|)$, where $d$ and $c$ denotes the average degree of graph and predefined community number. According to Eq.(11), the cost of meta-gradient computation is the same as the forward computation $O((d+1)c|\mathcal{V}|)$. Finally, the selection of optimal link would be $O(|\mathcal{V}| \log |\mathcal{V}|)$. Therefore, for a given node $v$, the total time complexity of suggesting a link to add would be $O(2(d+1)c|\mathcal{V}| + |\mathcal{V}| \log |\mathcal{V}|)$. In comparison, a GNN-based link recommendation has the time complexity as $O(dh|\mathcal{V}| + |\mathcal{V}| \log |\mathcal{V}|)$, where $h$ indicates the hidden dimension. Thus, the computational cost of FairGuide is similar to standard link recommendation methods. Additionally, the actual running time of adding a single link is reported in the Tab. 5 in the Appendix, which is less than 0.02 seconds.

## 5 Experiments

In this section, we conduct experiments on real-world user graphs to answer the following research questions:

- **RQ1**: Is FairGuide capable of adding new links that can effectively promote fairness in user graph structures?

- **RQ2**: How does the pseudo downstream task and dynamic link addition by meta-gradients contribute to FairGuide in guiding the fairness of graph structures ?

- **RQ3**: How does the number of new links introduced for fairness guidance influence the fairness of GNN-based applications?

## 5.1 Experimental Settings

### 5.1.1 Datasets

We adopt three real-world datasets for our experiments: Pokec-n, Pokec-z, and GitHub. The GitHub dataset is introduced in Sec. 3.2. The Pokec-z/n datasets Dai & Wang (2021) are both sampled from Pokec, which is the most popular social network in Slovakia. Node features include gender, age, hobbies, regions, and other attributes. Edges in the dataset represent friendship relationships among users. The statistics of three datasets are in the table 2. We set the labels larger than 1 as 1 and select 25% of labeled nodes as the validation set and 25% labeled nodes as the test set. We also select 7000 ground-truth labels as training set for GitHub dataset and 500 labels for Pokec datasets following the prior works.

Table 2: The statistics of the three datasets.

| Dataset | # Nodes | # Edges | Sensitive attribute | Label |
|---------|---------|---------|---------------------|-------|
| GitHub | 32,132 | 270,088 | Country | Popularity |
| Pokec-z | 67,797 | 882,765 | Region | Job Field |
| Pokec-n | 66,569 | 729,129 | Region | Job Field |

### 5.1.2 Baselines

Since our proposed framework guides graph fairness via link addition, we compare it with three pre-processing methods that could debias the graph structures via injecting new links. Additionally, we include two simple link-addition strategies, i.e., Rand. Add and Link Pred., for comparisons:

- **Rand. Add**: This baseline randomly add links to the graph.

- **Link Pred.** Schlichtkrull et al. (2018): It trains a GCN-based graph autoencoder. Then, pairwise cosine similarity between node embeddings are used for the link prediction.

- **EDITS** Dong et al. (2022): A preprocessing framework designed to mitigate bias in attributed networks by optimizing bias metrics across both attribute and structural modalities. We adopt its structure debiasing part and restrict it to only add edges according to the trained edge score.

- **Fairgen** Zheng et al. (2024): A generative model that promotes fairness in graph generation by incorporating label information and fairness constraints, thereby reducing representation disparity. We sample new edges in the generated graph structure and combine them with the original graph structure.

- **Graphair** Ling et al. (2023): A method for learning fair graph representations through the automatic discovery of fairness-aware augmentations.We choose its structure augmentation part and ensure the preservation of original edges.

### 5.1.3 Implementation Details

FairGuide is implemented using Pytorch and optimized via Adam optimizer Kingma & Ba (2014). Each iteration we fix the adding number as 100. For each method, we conduct experiments with seed {10,20,30,40,50}

Table 3: Results for the node classification on GCN model, with best results in bold and runner-up results in gray.

| Dataset | Metrics (%) | Vanilla | Rand. Add | Link Pred. | Edits | Graphair | Fairgen | FairGuide |
|---------|-------------|---------|-----------|------------|-------|----------|---------|-----------|
| GitHub | F1 ($\uparrow$) | $78.6 \pm 0.2$ | $78.0 \pm 0.1$ | $78.8 \pm 0.1$ | $\mathbf{78.6 \pm 0.1}$ | $77.5 \pm 0.2$ | $77.5 \pm 0.2$ | $77.8 \pm 0.1$ |
| | AUC ($\uparrow$) | $85.4 \pm 0.5$ | $84.3 \pm 0.1$ | $85.4 \pm 0.1$ | $\mathbf{84.8 \pm 0.3}$ | $84.2 \pm 0.1$ | $84.0 \pm 0.1$ | $84.3 \pm 0.1$ |
| | $\Delta_{SP}$ ($\downarrow$) | $12.5 \pm 0.4$ | $11.0 \pm 0.2$ | $11.9 \pm 0.2$ | $11.5 \pm 0.8$ | $11.1 \pm 0.3$ | $10.8 \pm 0.3$ | $\mathbf{8.6 \pm 0.2}$ |
| | $\Delta_{EO}$ ($\downarrow$) | $8.5 \pm 0.5$ | $8.3 \pm 0.3$ | $8.3 \pm 0.4$ | $8.5 \pm 0.8$ | $8.4 \pm 0.3$ | $8.3 \pm 0.1$ | $\mathbf{6.0 \pm 0.2}$ |
| pokec-n | F1 ($\uparrow$) | $66.8 \pm 0.7$ | $67.1 \pm 0.3$ | $66.7 \pm 0.5$ | $65.4 \pm 0.4$ | $65.4 \pm 0.3$ | $65.2 \pm 0.3$ | $\mathbf{66.2 \pm 0.3}$ |
| | AUC ($\uparrow$) | $75.4 \pm 0.1$ | $74.8 \pm 0.1$ | $75.4 \pm 0.2$ | $72.8 \pm 0.6$ | $74.3 \pm 0.1$ | $74.2 \pm 0.2$ | $\mathbf{74.9 \pm 0.2}$ |
| | $\Delta_{SP}$ ($\downarrow$) | $8.5 \pm 0.7$ | $9.2 \pm 1.0$ | $7.9 \pm 0.8$ | $3.7 \pm 1.3$ | $9.7 \pm 0.4$ | $9.6 \pm 0.7$ | $\mathbf{1.3 \pm 0.8}$ |
| | $\Delta_{EO}$ ($\downarrow$) | $11.9 \pm 1.4$ | $12.3 \pm 1.0$ | $11.0 \pm 1.6$ | $6.3 \pm 1.5$ | $11.7 \pm 1.6$ | $11.7 \pm 0.5$ | $\mathbf{3.1 \pm 0.5}$ |
| pokec-z | F1 ($\uparrow$) | $70.6 \pm 0.4$ | $70.5 \pm 0.2$ | $70.3 \pm 0.6$ | $67.1 \pm 1.1$ | $69.0 \pm 0.5$ | $69.1 \pm 0.9$ | $\mathbf{70.2 \pm 0.4}$ |
| | AUC ($\uparrow$) | $77.2 \pm 0.1$ | $76.8 \pm 0.1$ | $77.2 \pm 0.1$ | $75.5 \pm 0.7$ | $76.0 \pm 0.1$ | $76.0 \pm 0.3$ | $\mathbf{76.1 \pm 0.1}$ |
| | $\Delta_{SP}$ ($\downarrow$) | $9.1 \pm 0.9$ | $8.8 \pm 0.8$ | $9.5 \pm 0.8$ | $5.0 \pm 2.0$ | $6.4 \pm 1.5$ | $6.9 \pm 2.4$ | $\mathbf{3.1 \pm 0.7}$ |
| | $\Delta_{EO}$ ($\downarrow$) | $8.2 \pm 1.2$ | $8.0 \pm 0.7$ | $8.1 \pm 1.1$ | $5.1 \pm 1.8$ | $5.5 \pm 2.0$ | $7.6 \pm 2.6$ | $\mathbf{4.6 \pm 0.4}$ |

Table 4: Results on the downstream community detection task, with best results in bold and runner-up results in gray.

| Dataset | Metrics (%) | Vanilla | Rand. Add | Link Pred. | Edits | Graphair | Fairgen | FairGuide |
|---------|-------------|---------|-----------|------------|-------|----------|---------|-----------|
| GitHub | $\Delta_{SP}$ ($\downarrow$) | $41.9 \pm 1.5$ | $35.7 \pm 2.8$ | $38.2 \pm 4.0$ | $40.2 \pm 3.6$ | $31.9 \pm 2.1$ | $36.3 \pm 3.8$ | $\mathbf{29.0 \pm 2.2}$ |
| pokec-n | $\Delta_{SP}$ ($\downarrow$) | $83.2 \pm 3.0$ | $76.1 \pm 0.8$ | $77.5 \pm 2.6$ | $79.6 \pm 3.3$ | $79.2 \pm 2.9$ | $80.0 \pm 1.4$ | $\mathbf{74.1 \pm 3.4}$ |
| pokec-z | $\Delta_{SP}$ ($\downarrow$) | $77.7 \pm 0.9$ | $79.3 \pm 3.7$ | $80.4 \pm 3.0$ | $82.3 \pm 3.8$ | $81.7 \pm 1.7$ | $79.6 \pm 0.7$ | $\mathbf{70.1 \pm 4.0}$ |

and compute the mean and standard deviations of F1, AUC, $\Delta_{SP}$ and $\Delta_{EO}$. All methods consistently add the same number of new edges, corresponding to 3% of the original graph's total edge number for Pokec-n, 1.5% for Pokec-z and 4% for GitHub. For GNN backbone, we employ 2-layer GCN, 2-layer GraphSage and 10-layer APPNP. The hidden dimension is set as 128, the learning rate and training epochs are set as $1 \times 10^{-3}$ and 1000 respectively.

## 5.2 Results of Fairness Guidance

To answer **RQ1**, we first demonstrate that the links suggested by FairGuide effectively improve the fairness of various GNN models across different downstream tasks. Then, we discuss the trade-off between fairness and utility in user graphs.

**Fairness Improvements on Downstream Tasks**. Two types of downstream tasks, i.e., node classification and community detection, are utilized to evaluate the effectiveness of FairGuide in guiding the user graph structures toward fairness. For node classification task, a GCN is trained on these updated graph structures as the downstream classifier. For community detection task, we adopt the Louvain method Blondel et al. (2008) based on modularity optimization. We utilize the $\Delta_{SP}$ and $\Delta_{EO}$ as metrics to evaluate the fairness of the GCN-based downstream classifiers. Fairer results in downstream tasks indicate a better performance in guiding the graph structure towards fairness. The results on two downstream tasks are presented in Tab. 3 and Tab. 4. From these tables, we observe that:

- For the node classification task, all baselines perform poor in fairness metric. In contrast, models trained on graphs with new links added by FairGuide achieves great performance in fairness and only sacrifices little task performance.

- For the community detection task, the Rand. Add baseline shows limited effectiveness in enhancing fairness. Furthermore, links added by some baseline methods even amplify biases between communities. By contrast, fairness-guided links added by FairGuide could bring significant improvements in community detection fairness compared with baselines.

We also provide further analyses in Appendix C, including additional downstream tasks and the impact of added links on graphs.

**Comparison with fair link predictor**. Besides preprocessing baselines, We additionally compare against two fair link prediction methods: FairLp Li et al. (2022) and a within-group fairness regularization strategy WG-Reg Subramonian et al. (2024). We adopt these fair link predictors to add the same number of fair links as our FairGuide. As shown in Fig. 3(a), our FairGuide consistently achieves greater fairness improvements in downstream tasks compared to fair link prediction methods.

**Trade-off between Utility and Fairness**. We visualize the F1-$\Delta_{SP}$ of different methods by varying the strength of fairness guidance. Specifically, we vary the percentage of adding fairness-guided links from 0.5% to 3%. The results are given in Fig. 3(b–c). Scattered points in various colors represent results from different methods. We further fit a straight line for each method. From these two figures, we observe that great improvements in fairness are often accompanied with a decrease in task performance. Compared to other baseline models, FairGuide achieves a better utility-fairness balance. FairGuide either has better task performance at the same fairness level or reduce more bias at equivalent task performance.

**Generalization to Various GNN Architectures**. In addition, to further verify the structural fairness of graphs guided by FairGuide, we evaluate the impact of fairness-guided links on two different GNN backbones, i.e. , GraphSage Hamilton et al. (2017) and APPNP Gasteiger et al. (2019). Results in fairness are presented in Fig. 4(a–b). From figure, we observe that different GNN backbones consistently achieve fairness improvements when trained on graphs guided by FairGuide. This demonstrates that FairGuide can grow fair graph structures that facilitate different GNN models.

## 5.3 Ablation Study

FairGuide relies on two critical components to effectively guide the fairness of graph structures: the pseudo downstream task and link addition with Gumbel-max sampling. To assess the impact of these components to answer **RQ2**, we conduct ablation studies using the GCN-based node classification task.Specifically, to demonstrate the importance of employing community detection as the pseudo downstream task, we compare with a variant named FairGuide\C where the initial community labels are replaced with randomly generated labels. Additionally, to evaluate the effectiveness of Gumbel-max sampling in the fairness-guided link addition , we introduce another variant, FairGuide\S, in which links are determined through a single iteration without dynamic updates. The results are shown in Fig. 4(c–d), where we observe:

- FairGuide achieves smaller $\Delta_{SP}$ compared to the variant FairGuide\C, where the community detection labels are replaced by propagated random labels. This is because community labels correlate with labels in downstream tasks. Thus, incorporating community detection as a pseudo task effectively enhances fairness in downstream tasks, which empirically verifies our theoretical analysis.

- FairGuide\S consistently underperforms FairGuide in fairness evaluations across all datasets, thereby highlighting the importance of dynamic link addition strategies via Gumbel-max sampling.

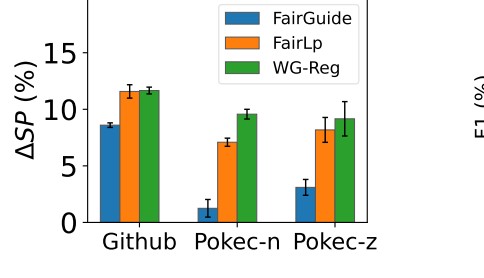
(a) Compared with fair link predictor.

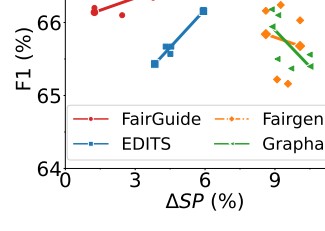
(b) Trade-off on Pokec-n.

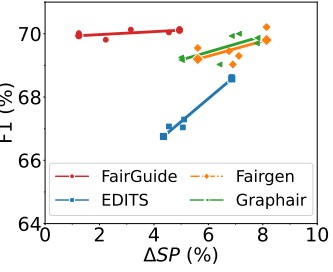
(c) Trade-off on Pokec-z.

Figure 3: (a) Comparison with fair link predictors in guiding the graph structures toward fairness. (b–c) Visualization of utility-fairness trade-off; methods in the upper-left indicate better utility and fairness.

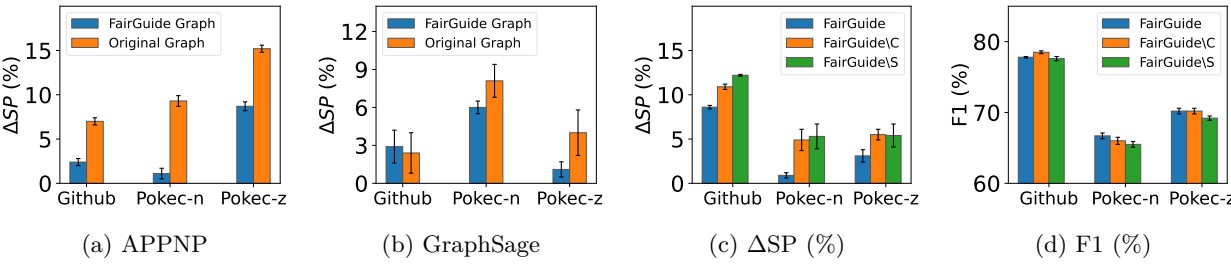

(a) APPNP      (b) GraphSage      (c) ΔSP (%)      (d) F1 (%)

Figure 4: (a–b) Fairness improvements across GNN backbones. (c–d) Ablation studies on three user graphs.

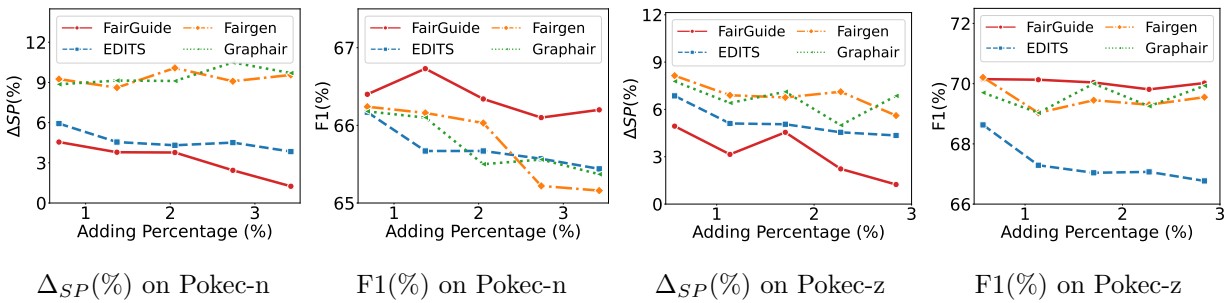

$\Delta_{SP}(\%)$ on Pokec-n     F1(%) on Pokec-n     $\Delta_{SP}(\%)$ on Pokec-z     F1(%) on Pokec-z

Figure 5: Impacts of number of added links.

## 5.4 Impacts of the Number of New Links

To answer the **RQ3**, we vary edge adding percentage based on the existing edges from 0.5% to 3% to investigate the impact of link addition size. We conduct experiments for node classification task based on GCN. The impacts of link addition rate on pokec-n and pokec-z are shown in Fig. 5. We have similar observations on other datasets. From these results, we can observe that with FairGuide, the fairness of the downstream GNN classifier improves as the percentage of added edges increases. During this process, FairGuide consistently outperforms baseline methods in terms of fairness while preserving utility, which demonstrates the effectiveness of FairGuide in guiding graph structures toward fairness.

## 5.5 Hyperparameter Analysis

In FairGuide, the cross-group boost rate $\beta$ and the number of community clusters C are two key hyperparameters.To investigate how the two hyperparameters affect fairness and task performance, We conduct GCN node classification task on the Pokec-n. Similar trends are also observed in other datasets. We vary $\beta$ to $\{0.01,0.1,1.0,10.0,100.0\}$ and $C$ to $\{5,10,15,20,25\}$, respectively. All other settings follow the description given in Sec. 5.1. As shown in Fig. 6, the task performance F1 score remains stable across the tested hyperparameter ranges, fluctuating by only around 1–2 percentage points.

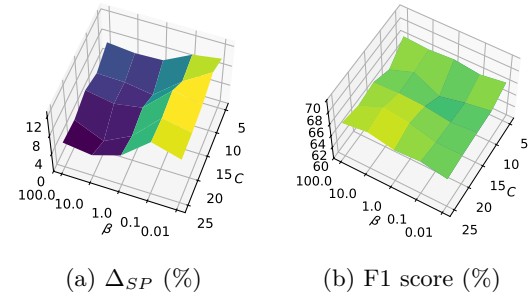

(a) $\Delta_{SP}$ (%)      (b) F1 score (%)

Figure 6: Hyperparameter sensitivity analysis.

For the fairness performance, FairGuide achieves a strong debiasing effect when $C \geq 15$ and $\beta \geq 1.0$. A larger number of clusters $C$ allows the model to capture more fine-grained feature information, and a large cross-group boost rate $\beta$ effectively establishes connections between different sensitive groups, which will benefit the graph structural fairness.

## 6 Conclusion and Future Work

In this work, we study a novel problem of fairness guidance via link addition which aims to guide existing graph structures toward fairness. To solve this problem, we form it as bi-level optimization problem and

propose an algorithm called FairGuide which utilizes the differentiable pseudo community detection task to optimize the graph structure. Specifically, we theoretically demonstrate generalize fairness can be achieved by reducing the bias in pseudo task via new links. To efficiently identify beneficial new links, we compute meta-gradients with respect to the graph structure and adopt an edge-sampling technique to discretely update the structure. Experiment results on real-world datasets show the effectiveness of our proposed algorithm in solving the fair guidance problem. For the future work, there are several directions to investigate. First, in this paper links are added just according to the fairness without consideration of enhancing the structure utility. In future we will explore how to add links to simultaneously improve the task performance and fairness. Second, solving various sensitive attribute by adding links is also an interesting problem. We plan to extend the FairGuide to achieve more general fairness on diverse sensitive attributes.

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

## Appendix A   Algorithm of FairGuide

As shown in Algorithm 1, FairGuide first initializes community labels by K-means and achieve differential community detection. Then, FairGuide further computes the $\Delta_{SP}$ metric and meta-gradient to the graph structure. Finally, new links sampled based on meta-gradients of structure are selected.

---

**Algorithm 1:** Algorithm of FairGuide

---

**Input:** $A$: adjacency matrix; $X$: node feature; $\Delta$: link addition constraint; $K$: number of aggregation steps; $\alpha$: restart parameter; $S$: sensitive attribute vector
**Output:** Fairness-guided graph structure $\mathbf{A}'$ with new links
$\mathbf{A}' \leftarrow \mathbf{A}$;
$\mathbf{C_{init}} \leftarrow$ K-Means($f_{\mathrm{MLP}}(X)$)
**while** *number of added links do not reach $\Delta$* **do**
    **Differential Community Detection:**
    Obtains the community labels $\mathbf{C^{(K)}}$ by Eq.(9)
    **Compute Fairness Loss:**
    $\Delta_{SP} \leftarrow \Delta_{SP}(\mathbf{C^{(K)}}, \mathcal{S})$
    **Compute Edge Score:**
    $\nabla_{\mathbf{A}'}^{\mathrm{meta}} \leftarrow -\frac{\partial \Delta_{SP}}{\partial \mathbf{C}} \frac{\partial \mathbf{C}}{\partial \mathbf{A}'} \odot (1 + \beta \cdot \mathbb{I}(s_i \neq s_j))$ ;
    **Gumbel Edge Sampling:**
    $G_{i,j} \sim \mathrm{Gumbel}(0,1)$
    $\mathbf{W} = \log(-\nabla_{\mathbf{A}'}^{\mathrm{meta}} \odot (\mathbf{A}' == 0)$
    $\mathbf{P} = \frac{\log(\mathbf{W}+\epsilon)+G_{i,j}}{\tau}$ , $\delta \leftarrow \mathrm{Top}_k(\mathbf{P})$
    **Batch Edge Addition:**
    $\mathbf{A}' \leftarrow \mathbf{A}' + \sum_{(i,j)\in\delta} \mathbf{E}_{i,j}$
**return** $\mathbf{A}'$;

---

Table 5: Running time of FairGuide for adding one link.

| Dataset | Github | Pokec-n | Pokec-z |
|---|---|---|---|
| Run Time | $4.6 \times 10^{-3}$s | $1.6 \times 10^{-2}$s | $1.2 \times 10^{-2}$s |

## Appendix B   Proof of Theorem 1

*Proof.* To prove this theorem, we first introduce the following lemmas and theorem:

**Lemma 1.** *Given a unit sphere centered at origin $O$, let $A, B, C$ be three points on the surface. Assume angles $AOB = \theta_1$ and $BOC = \theta_2$. Then, the cosine of angle $AOC$ satisfies:*

$$\cos(AOC) \in [\cos(\theta_1 + \theta_2), \cos(\theta_1 - \theta_2)].$$

*Proof.* From the spherical law of cosines:

$$\cos \theta_3 = \cos \theta_1 \cos \theta_2 + \sin \theta_1 \sin \theta_2 \cos B',$$

where $B'$ is the angle opposite to $B$ in the spherical triangle $ABC$. Since angles are in $[0, \pi]$, we have:

$$\cos \theta_3 \geq \cos \theta_1 \cos \theta_2 - \sin \theta_1 \sin \theta_2 = \cos(\theta_1 + \theta_2),$$
$$\cos \theta_3 \leq \cos \theta_1 \cos \theta_2 + \sin \theta_1 \sin \theta_2 = \cos(\theta_1 - \theta_2).$$

Thus, the lemma is proven. □

**Lemma 2.** *Given two random variables $X$ and $Y$, the Pearson correlation coefficient is equivalent to the cosine similarity between their z-score vectors $x'$ and $y'$, where:*

$$x'_i = \frac{X_i - \mu_X}{\sigma_X}, \quad y'_i = \frac{Y_i - \mu_Y}{\sigma_Y}.$$

*Proof.* By definition, the Pearson correlation coefficient is:

$$\rho_{X,Y} = \frac{\mathbb{E}[(X - \mu_X)(Y - \mu_Y)]}{\sigma_X \sigma_Y} = \lim_{n \to \infty} \frac{1}{n} \sum_{i=1}^{n} x'_i y'_i = \cos(x', y').$$

Thus, the lemma is proven. $\square$

**Theorem 2.** *Given three random variables $X, Y, Z$ with correlation coefficients $\rho_{X,Y} = \cos\alpha$ and $\rho_{Y,Z} = \cos\beta$, then:*

$$\rho_{X,Z} \in [\cos(\alpha + \beta), \cos(\alpha - \beta)].$$

*Proof.* From Lemma 2, the correlation coefficients correspond to cosine similarities between z-score vectors. Applying Lemma 1 with these vectors yields:

$$\rho_{X,Z} = \cos(x', z') \in [\cos(\alpha + \beta), \cos(\alpha - \beta)].$$

Thus, the theorem is proven. $\square$

Now, applying Theorem 2 directly, we have:

Given $\rho_{C,\hat{Y}} = \cos\alpha$, and $\rho_{S,C}$ within $[\frac{\pi}{2} + \delta, \frac{\pi}{2} - \delta]$, we obtain:

$$\rho_{S,\hat{Y}} \in \left[\cos\left(\frac{\pi}{2} + \delta + \alpha\right), \cos\left(\frac{\pi}{2} - \delta - \alpha\right)\right].$$

Thus, the theorem is proven. $\square$

## B.1 Connection between Correlation and Statistical Parity

Theorem 1 is stated with Pearson correlation, while the objective in Eq. (6) uses statistical parity over community assignments. In the binary case, these two quantities are directly related because Pearson correlation reduces to the Phi coefficient. For a binary community variable $C$, we have:

$$|\rho(S, C)| = |P(C = 1 \mid S = 1) - P(C = 1 \mid S = 0)| \sqrt{\frac{P(S = 1)P(S = 0)}{P(C = 1)P(C = 0)}}. \tag{16}$$

The first term on the right-hand side is exactly the statistical-parity gap:

$$\Delta_{SP}(C, S) = |P(C = 1 \mid S = 1) - P(C = 1 \mid S = 0)|. \tag{17}$$

Therefore, in the binary setting, the Pearson-correlation objective and the statistical-parity objective are linked by a marginal normalization factor. Extending this connection to multi-class community labels is a meaningful theoretical direction, we leave this extension as future work.

## Appendix C    Additional Analyses

### C.1    Heuristic Link-addition Baselines

We further include two representative heuristic link-addition baselines for a more complete comparison. Table 6 reports Cross-group Random and Similarity-constrained Cross-group, both adding the same number of links as FairGuide. Compared with these heuristics, FairGuide achieves lower fairness gaps on all datasets.

Table 6: Comparison with representative heuristic link-addition baselines under GCN node classification.

| Dataset | Cross-group Random | | Similarity-constrained | | FairGuide | |
|---|---|---|---|---|---|---|
| | $\Delta_{SP}$ (%) $\downarrow$ | $\Delta_{EO}$ (%) $\downarrow$ | $\Delta_{SP}$ (%) $\downarrow$ | $\Delta_{EO}$ (%) $\downarrow$ | $\Delta_{SP}$ (%) $\downarrow$ | $\Delta_{EO}$ (%) $\downarrow$ |
| GitHub | $9.3 \pm 0.5$ | $7.1 \pm 0.5$ | $9.9 \pm 0.3$ | $7.4 \pm 0.3$ | $\mathbf{8.6 \pm 0.2}$ | $\mathbf{6.0 \pm 0.2}$ |
| pokec-n | $5.1 \pm 1.4$ | $9.0 \pm 1.5$ | $8.5 \pm 0.9$ | $12.1 \pm 1.4$ | $\mathbf{1.3 \pm 0.8}$ | $\mathbf{3.1 \pm 0.5}$ |
| pokec-z | $8.5 \pm 1.7$ | $7.8 \pm 1.8$ | $9.6 \pm 0.8$ | $8.7 \pm 1.1$ | $\mathbf{3.1 \pm 0.7}$ | $\mathbf{4.6 \pm 0.4}$ |

## C.2 Pseudo-task Correlation Analysis

We compute the absolute Pearson correlation between the propagated pseudo-community assignment $\mathbf{C}$ used by FairGuide and the downstream prediction $\hat{Y}$ on the original graph. Table 7 reports this correlation together with the downstream fairness metrics after applying FairGuide. Even on Pokec-z, where the correlation is only 0.160, FairGuide still noticeably reduces downstream fairness gaps. This supports that the pseudo task can provide a useful fairness-guidance signal without requiring strong task-specific alignment.

Table 7: Pseudo-task correlation and downstream fairness after FairGuide.

| Dataset | $|\rho(\mathbf{C}, \hat{Y})|$ | $\Delta_{SP}$ (%) $\downarrow$ | $\Delta_{EO}$ (%) $\downarrow$ |
|---|---|---|---|
| GitHub | 0.247 | $8.6 \pm 0.2$ | $6.0 \pm 0.2$ |
| Pokec-n | 0.259 | $1.3 \pm 0.8$ | $3.1 \pm 0.5$ |
| Pokec-z | 0.160 | $3.1 \pm 0.7$ | $4.6 \pm 0.4$ |

## C.3 Generalization Beyond the Original Binary Labels

Since FairGuide guides the graph structure before a specific downstream task is fixed, its fairness effect should not be limited to the original binary labels used in the main experiments. We therefore evaluate additional downstream objectives on the fairness-guided graphs, including extra node-classification labels and a regression target. For classification, we keep the same sensitive attributes and replace the prediction label with another user attribute. As shown in Table 8, FairGuide reduces fairness gaps for these additional downstream labels.

Table 8: Additional downstream-label experiments.

| Dataset | New target | Graph | $\Delta_{SP}$ (%) $\downarrow$ | $\Delta_{EO}$ (%) $\downarrow$ |
|---|---|---|---|---|
| pokec-n | `I_like_music_indicator` | Original | $12.4 \pm 2.7$ | $7.7 \pm 2.0$ |
| pokec-n | `I_like_music_indicator` | FairGuide | $\mathbf{5.5 \pm 3.3}$ | $\mathbf{2.7 \pm 2.4}$ |
| GitHub | `public_repos` | Original | $13.4 \pm 1.6$ | $17.7 \pm 2.5$ |
| GitHub | `public_repos` | FairGuide | $\mathbf{10.9 \pm 0.8}$ | $\mathbf{15.6 \pm 1.4}$ |

For regression task, we evaluate a GitHub follower-count task where the model directly predicts raw follower count instead of a binarized popularity label. Following fair-regression notions of statistical parity and group-loss/error parity (Agarwal et al., 2019; Chzhen et al., 2020; Gursoy & Kakadiaris, 2022), we evaluate prediction-distribution disparity using Wasserstein-1 and Kolmogorov-Smirnov distances, and error disparity using group MAE gap and absolute-error distribution distances. Table 9 shows that FairGuide reduces both prediction-distribution disparity and error-distribution disparity across sensitive groups.

## C.4 Graph Structural Integrity and Link Prediction

We measure how the added links affect graph structural properties. Table 10 reports sensitive homophily, label homophily, Louvain modularity, and largest-community ratio before and after applying FairGuide. FairGuide reduces sensitive homophily, consistent with improving fairness-relevant structural exposure. At

Table 9: Fairness metrics for GitHub follower-count regression.

| Graph | Group MAE gap↓ | Prediction W1↓ | Prediction KS↓ | Abs. error W1↓ | Abs. error KS↓ |
|---|---|---|---|---|---|
| Original | 45.4±5.9 | 80.7±5.5 | 0.16±0.01 | 61.4±5.6 | 0.13±0.01 |
| FairGuide | **5.2±0.5** | **51.1±0.6** | **0.10±0.01** | **14.7±0.5** | **0.02±0.01** |

the same time, label homophily changes only slightly, modularity decreases moderately, and the largest-community ratio does not increase.

Table 10: Graph structural properties before and after FairGuide.

| Dataset | Graph | Sensitive homophily | Label homophily | Modularity | Largest community ratio |
|---|---|---|---|---|---|
| GitHub | Original | 0.661 | 0.765 | 0.474 | 0.248 |
| GitHub | FairGuide | 0.613 | 0.743 | 0.456 | 0.211 |
| Pokec-n | Original | 0.953 | 0.591 | 0.567 | 0.271 |
| Pokec-n | FairGuide | 0.870 | 0.587 | 0.547 | 0.242 |
| Pokec-z | Original | 0.951 | 0.587 | 0.574 | 0.288 |
| Pokec-z | FairGuide | 0.921 | 0.586 | 0.564 | 0.266 |

We further evaluate link prediction on held-out original edges to examine whether the added links preserve useful topological information. The positive validation/test edges are always sampled from the original graph; FairGuide-added links are used only as message-passing topology and are not treated as positive labels. Table 11 shows comparable link-prediction performance after applying FairGuide.

Table 11: Link prediction on held-out original edges.

| Dataset | Graph | AUC↑ | AP↑ |
|---|---|---|---|
| GitHub | Original | 0.865±0.013 | 0.874±0.014 |
| GitHub | FairGuide | 0.852±0.018 | 0.866±0.018 |
| Pokec-n | Original | 0.757±0.033 | 0.722±0.029 |
| Pokec-n | FairGuide | 0.760±0.032 | 0.758±0.032 |

### C.5 Semantic Plausibility of Added Links

Following recent uses of LLMs for role-play simulation and evaluation (Park et al., 2023; Zheng et al., 2023), we evaluate whether FairGuide-added links are plausible as user-facing recommendations. On 500 GitHub links, a role-playing LLM judges 301 links as semantically meaningful.

### C.6 Role-play Prompt

We use the following prompt template for the role-play evaluation. The profile fields are filled with the visible raw-profile information of User U and the recommended User V.

**System prompt.**

```
You are role-playing a GitHub user who receives a connection recommendation in a
    developer network.
Make the decision from the perspective of User U, based on the visible profile
    fields of User U and the recommended user.
Return strict JSON with keys:
rank, semantically_meaningful, likely_considered_by_user, score, rationale.
score must be from 1 to 5.
```

**User prompt template.**

```
You are User U in a developer network.
A recommendation system suggests User V as a potential connection.

Your profile:
<User U profile JSON>

Recommended user's profile:
<User V profile JSON>

Question:
As User U, would you consider connecting with or following User V?
Is this recommendation meaningful for you in a developer-network setting?
Return strict JSON only.
```

## Appendix D    Semantic Plausibility Case Studies

To examine whether the added links can be interpreted as meaningful recommendations, we conduct a role-play evaluation on 500 FairGuide-added GitHub links. The LLM is asked to act as the source user and judge whether the recommended user is meaningful for connection based on visible profile information. Overall, 301/500 links are judged semantically meaningful. Table 12 lists representative anonymized cases.

Table 12: Representative anonymized raw-profile case studies for FairGuide-added GitHub links.

| Case | Groups | Profile compatibility | Activity comparison | Role-play judgment |
|---|---|---|---|---|
| A | 1 / 0 | Both are JavaScript developers. One profile is associated with a small development organization and the other with independent/freelance development. | repos 34/29; followers 14/13; following 5/6; gists 0/0 | meaningful, likely considered, score 4 |
| B | 0 / 1 | Both are JavaScript/app-development profiles. Both profile texts indicate app/programming-related work. | repos 47/51; followers 10/20; following 8/8; gists 1/2 | meaningful, likely considered, score 4 |
| C | 0 / 1 | Both are Python profiles. One user is associated with an industry technology organization and the other with an academic CS profile. | repos 37/56; followers 15/18; following 15/21; gists 0/0 | meaningful, likely considered, score 4 |
| D | 0 / 1 | Both use HTML. One profile indicates senior software-engineering and web/frontend experience, matching the other user's developer profile. | repos 41/35; followers 15/12; following 36/37; gists 0/1 | meaningful, likely considered, score 4 |
| E | 0 / 1 | Both are Java profiles with equal repository counts. One profile indicates enterprise Java development and the other is associated with Android-related development. | repos 85/85; followers 33/28; following 14/24; gists 0/6 | meaningful, likely considered, score 4 |

## Broader Impact Statement

There is no ethical issue in the preparation of this work. All datasets used in this study containing sensitive information were collected from publicly available information and have undergone anonymization processing.

