# OpenReview forum: "Let’s Grow an Unbiased Community : Guiding the Fairness of Graphs via New Links"
_TMLR — Under review for TMLR_

### Review · Reviewer_8mvF · 2026-06-11

**Summary Of Contributions:**

Summary:
This paper proposes FairGuide, a graph preprocessing framework that adds a limited number of new links to biased user graphs in order to improve group fairness. It uses differentiable community detection as a task-agnostic proxy and selects fairness-guided links via meta-gradients and Gumbel-max sampling.

Strengths:

1. The paper studies a relatively interesting link-addition-only setting for graph fairness, which is different from standard fair GNN training.
2. The proposed framework is evaluated on multiple datasets and GNN backbones, with ablation studies showing the contribution of the pseudo community task and dynamic link addition.

Weaknesses:

1. The practical motivation is not fully convincing. The paper assumes that adding new links can guide real user graphs toward fairness, but it does not validate whether the suggested links are realistic, semantically meaningful, or likely to be accepted by users.
2. The usefulness of the proposed problem setting is unclear beyond mutable social/user graphs. For many graph domains, adding edges is not a valid intervention.
3. The task-agnostic claim relies heavily on community detection as a pseudo task. The assumption that community fairness transfers to diverse downstream tasks is strong and may not hold when downstream labels are weakly related to community structure.
4. The baseline comparison is not fully convincing. Stronger heuristic baselines such as cross-group random addition, degree-aware cross-group addition, community-bridge link addition, and similarity-constrained cross-group addition are missing.
5. The evaluation mainly demonstrates offline improvements in fairness metrics after modifying the graph. It does not evaluate link plausibility, long-term graph evolution, user acceptance, or whether the added links preserve meaningful social relations.

**Audience:**

Yes

**Audience Explanation:**

The topic is relevant to graph fairness and fair graph preprocessing.

**Claims And Evidence:**

No

**Claims Explanation:**

The paper shows some fairness improvements on the evaluated datasets, but the evidence is not fully convincing for the broader claims. In particular, it does not validate whether the added links are realistic or meaningful in real user graphs. The task-agnostic claim also mainly relies on community detection as a proxy task, with limited downstream evaluation. In addition, stronger link-addition heuristic baselines are missing.

**Requested Changes:**

Please see the weaknesses above.

---

> ### Author Response · Authors · 2026-07-12
>
> We thank the reviewer for the constructive comments on the practical meaning of added links, the scope of the problem setting, the pseudo-task assumption, and the need for stronger link-addition comparisons.
>
>
> ### Response to Weaknesses 1 & 5: Meaningfulness of Added Links
>
> To further evaluate whether FairGuide-added links are plausible as user-facing recommendations, we added a semantic plausibility analysis on GitHub. Following recent role-play simulation and LLM-as-judge studies (Park et al., 2023; Zheng et al., 2023), we construct 500 FairGuide-added link cases and ask an LLM to role-play the source user. The prompt provides visible raw-profile information of the source user and the recommended user, including profile text, location group, company/organization summary, primary programming language, and activity statistics. The LLM then returns a structured judgment on whether the recommended link is semantically meaningful.
>
> As reported in the revised appendix, 301/500 FairGuide-added links are judged semantically meaningful. We also add representative case studies, where the added links commonly connect users with compatible technical interests, programming languages, profile descriptions, or activity levels, suggesting that the recommended links generally make sense as developer-network connections.
>
>
> ### Response to Weaknesses 1 & 2: Motivation and Applicability Scope
>
> Our setting targets fairness-relevant user graphs, where edges represent social, collaborative, or interaction relations. In this scope, link addition is a practical intervention because a platform can recommend new connections while preserving all existing links and user attributes, with the amount of change controlled by the edge budget. Graph domains where edge addition is not meaningful, such as molecular or protein graphs, are also outside the fairness scope considered in this work.
>
>
> ### Response to Weakness 3: Pseudo Task and Downstream Fairness
>
> We agree that the relationship between the pseudo task and downstream fairness should be supported more explicitly. Our theoretical analysis does not require the pseudo task to be identical to the downstream task. It only requires a non-trivial relation between the pseudo-community signal $C$ and the downstream prediction $\hat{Y}$; then reducing the dependence between the sensitive attribute $S$ and $C$ can constrain the upper bound of the dependence between $S$ and $\hat{Y}$.
>
> Empirically, we further study this relation on all three datasets. We report the absolute Pearson correlation between the pseudo-community signal and the downstream prediction on the original graph, together with the downstream fairness metrics after applying FairGuide:
>
> | Dataset | $\lvert\rho(C,\hat{Y})\rvert$ |  $\Delta_{SP}$ (%) ↓ | $\Delta_{EO}$ (%) ↓ |
> |---|---:|---:|---:|
> | GitHub | 0.247 | 8.6 ± 0.2 | 6.0 ± 0.2 |
> | Pokec-n | 0.259 | 1.3 ± 0.8 | 3.1 ± 0.5 |
> | Pokec-z | 0.160 | 3.1 ± 0.7 | 4.6 ± 0.4 |
>
> Even on Pokec-z, where the correlation is only 0.160, FairGuide still noticeably reduces the downstream fairness metrics, supporting that the pseudo task can help constrain downstream fairness disparity without strong task-specific alignment.
>
> ### Response to Weakness 4: Additional Heuristic Link-Addition Baselines
>
> For a more complete comparison, we added two heuristic link-addition baselines, Cross-group Random and Similarity-constrained Cross-group link addition. Under the node classification task, FairGuide still obtains lower fairness gaps on all three datasets:
>
> | Dataset | Cross-group Random $\Delta_{SP}$ (%) ↓ | Cross-group Random $\Delta_{EO}$ (%) ↓ | Similarity-constrained $\Delta_{SP}$ (%) ↓ | Similarity-constrained $\Delta_{EO}$ (%) ↓ | FairGuide $\Delta_{SP}$ (%) ↓ | FairGuide $\Delta_{EO}$ (%) ↓ |
> |---|---:|---:|---:|---:|---:|---:|
> | GitHub | 9.3 $\pm$ 0.5 | 7.1 $\pm$ 0.5 | 9.9 $\pm$ 0.3 | 7.4 $\pm$ 0.3 | **8.6 $\pm$ 0.2** | **6.0 $\pm$ 0.2** |
> | Pokec-n | 5.1 $\pm$ 1.4 | 9.0 $\pm$ 1.5 | 8.5 $\pm$ 0.9 | 12.1 $\pm$ 1.4 | **1.3 $\pm$ 0.8** | **3.1 $\pm$ 0.5** |
> | Pokec-z | 8.5 $\pm$ 1.7 | 7.8 $\pm$ 1.8 | 9.6 $\pm$ 0.8 | 8.7 $\pm$ 1.1 | **3.1 $\pm$ 0.7** | **4.6 $\pm$ 0.4** |
>
> These results further support the importance of fairness-guided candidate-edge scoring.

---

### Review · Reviewer_ojA3 · 2026-06-21

**Summary Of Contributions:**

This paper introduces FairGuide, a framework for fairness guidance in graphs that adds a limited number of new links to a graph in order to reduce bias in downstream tasks. The graph fairness guidance problem is framed in the bilevel optimization formulation and the authors propose to replace the downstream task with a pseudo community detection task. To enable gradient-based optimization of the fairness objective over the graph structure, the paper presents a two-stage differentiable community detection method that decouples the task into graph structure-irrelevant clustering and structure-based aggregation. The algorithm then computes the gradient of statistical parity with respect to the adjacency matrix and uses Gumbel-max sampling, guided by this gradient, to select links for addition. Empirical results on three real-world datasets show that the proposed method outperforms several baselines on fairness guidance while achieving a better trade-off between downstream-task performance and fairness.

**Strengths**

1. The paper focuses on an important problem, fairness guidance in graphs, which has great application value.
2. The performance of the proposed method is well-demonstrated in the main experiments, along with the supporting ablation studies and analyses.

**Weaknesses**

1. The authors claim their method is downstream task-agnostic, but both the empirical and theoretical evidence for this claim are limited. (See Weakness 2 and the section below for details.)
2. The theoretical justification is indirect. In Theorem 1, the fairness metric is expressed in terms of the Pearson correlation coefficient, but in practice (Equation 6), the statistical parity $\Delta_{\mathrm{SP}}(\mathbf{C}, \mathcal{S})$ is used, the paper does not address the potential gap between these two quantities. In addition, some steps in the proof lack rigor. In Lemma 2, it should be $\rho_{X, Y}=\lim_{n \to \infty}\frac{\sum_{i=1}^n x_i' y_i'}{n}$ rather than $\rho_{X, Y}=\lim_{n \to \infty}\sum_{i=1}^n x_i' y_i'$.
3. Some details of the algorithm are not clearly explained in the paper. It is only mentioned that the authors "utilize an MLP-based self-supervised auto-encoder to obtain latent feature representation for nodes," but what loss is used in training this MLP is not described clearly in the paper.

**Audience:**

Yes

**Audience Explanation:**

The paper focuses on fairness guidance in graphs, a problem with significant real-world applications.

**Broader Impact Concerns:**

None.

**Claims And Evidence:**

Yes

**Claims Explanation:**

The claims are partially supported.

As noted in Weakness 1, the method is claimed to be downstream task-agnostic, yet the experiments include only two downstream tasks, one of which (community detection) is the exact task the method is trained with. It therefore remains unclear whether the results generalize to other graph-based tasks. The theoretical results in Theorem 1 are relevant but do not directly support the claim that community detection is a suitable pseudo-task for fairness guidance (see Weakness 2).

On the positive side, the performance of the proposed framework is well demonstrated through the experimental comparisons against the baselines.

**Requested Changes:**

1. (Critical) The authors should include experiments on additional downstream tasks to support the claims: that using the community detection pseudo-task in the framework can improve fairness across diverse downstream tasks, and that the proposed method is indeed task-agnostic.
2. The authors state that "great improvements in fairness are always accompanied by a decrease in task performance." However, this does not hold for Fairgen and Graphair on Pokec-n, as shown in Figure 3(b). The paper should explain what might cause this phenomenon.
3. Minor issue. The sentence "Three real-world include Pokec-n, Pokec-z and GitHub are adopted for our experiments." should be revised.

Also see the weaknesses mentioned above.

---

> ### Author Response · Authors · 2026-07-12
>
> We thank the reviewer for the positive assessment of the problem and for the detailed comments on task-agnostic evidence, the theory-objective connection, and implementation details.
>
> ### Response to Weakness 1: Task-Agnostic Evidence
>
> We understand the reviewer's concern that the task-agnostic claim should be supported by diverse downstream evaluations. We have clarified that the original experiments already cover multiple GNN backbones for node classification, including GCN, GraphSAGE, and APPNP, as well as an unsupervised community-detection task. To further strengthen this evidence, we added experiments with additional downstream objectives. For classification, we keep the same sensitive attributes and replace the downstream label with another user attribute:
>
> | Dataset | New Target | Graph | $\Delta_{SP}$ (%) ↓ | $\Delta_{EO}$ (%) ↓ |
> |---|---|---|---:|---:|
> | Pokec-n | `I_like_music_indicator` | Original | 12.4 $\pm$ 2.7 | 7.7 $\pm$ 2.0 |
> | Pokec-n | `I_like_music_indicator` | FairGuide | **5.5 $\pm$ 3.3** | **2.7 $\pm$ 2.4** |
> | GitHub | `public_repos` | Original | 13.4 $\pm$ 1.6 | 17.7 $\pm$ 2.5 |
> | GitHub | `public_repos` | FairGuide | **10.9 $\pm$ 0.8** | **15.6 $\pm$ 1.4** |
>
> We also added a GitHub follower-count regression task, where the model predicts follower count of a developer. Following fair-regression notions of statistical parity and group-loss/error parity (Agarwal et al., 2019; Chzhen et al., 2020; Gursoy and Kakadiaris, 2022), we evaluate prediction-distribution disparity and error-distribution disparity:
>
> | Graph | Group MAE Gap ↓ | Prediction W1 ↓ | Prediction KS ↓ | Abs. Error W1 ↓ | Abs. Error KS ↓ |
> |---|---:|---:|---:|---:|---:|
> | Original | 45.4 $\pm$ 5.9 | 80.7 $\pm$ 5.5 | 0.16 $\pm$ 0.01 | 61.4 $\pm$ 5.6 | 0.13 $\pm$ 0.01 |
> | FairGuide | **5.2 $\pm$ 0.5** | **51.1 $\pm$ 0.6** | **0.10 $\pm$ 0.01** | **14.7 $\pm$ 0.5** | **0.02 $\pm$ 0.01** |
>
> These additional classification and regression experiments provide stronger evidence that FairGuide generalizes well across different downstream objectives.
>
> ### Response to Weakness 2: Connection Between Pearson Correlation and Statistical Parity
>
> We thank the reviewer for pointing out the gap between the correlation-based theorem and the statistical-parity objective. We revised the discussion to clarify this connection in the binary case, where Pearson correlation has a direct fairness interpretation.
>
> When both variables are binary, Pearson correlation reduces to the Phi coefficient. For a binary community variable $C$, we have:
>
> $$
> |\rho(S,C)| =
> \left|P(C=1 \mid S=1)-P(C=1 \mid S=0)\right|
> \sqrt{\frac{P(S=1)P(S=0)}{P(C=1)P(C=0)}} .
> $$
>
> The first term on the right-hand side is exactly the statistical-parity gap:
>
> $$
> \Delta_{SP}(C,S)=\left|P(C=1 \mid S=1)-P(C=1 \mid S=0)\right|.
> $$
>
> Therefore, in the binary setting, the Pearson-correlation objective and the statistical-parity objective are linked by a marginal normalization factor. This fills the gap between the theorem and the optimized fairness objective for binary downstream predictions. Extending this connection to multi-class community labels is a meaningful theoretical direction, we will discuss this as future work.
>
> ### Response to Weakness 3: Lemma 2 and MLP Auto-Encoder Details
>
> We corrected Lemma 2 by adding the missing normalization factor:
>
> $$
> \rho_{X,Y} = \lim_{n \to \infty} \frac{1}{n}\sum_{i=1}^{n} x'_i y'_i = \cos(x',y').
> $$
>
> We also added the missing implementation detail for the MLP module. The MLP is trained as a self-supervised auto-encoder: the encoder maps node features to latent representations, the decoder reconstructs the original node features, and the reconstruction objective is mean squared error.
>
> ### Response to other Requested Changes: Statement and Writing Issue
>
> We appreciate the reviewer pointing out this detail of the fairness-utility relationship. In the case of Pokec-n, FairGen and Graphair introduce only marginal fairness improvements., while their added edges can still introduce small task-performance fluctuations. Thus, the observed utility change in this case should be interpreted as graph-editing noise rather than a fairness-utility trade-off. We have also corrected the dataset sentence noted by the reviewer.

---

### Review · Reviewer_CNvd · 2026-06-29

**Summary Of Contributions:**

## **Strengths**

1. Addressing fairness by directly modifying the graph topology (adding links) is an interesting and distinct approach. It tackles the root cause of bias within the data structure itself, rather than adjusting the model or the output.
2. The introduction of a differentiable community detection task as a pseudo-downstream task is a strong design choice. By decoupling the fairness optimization from specific downstream labels, the method significantly improves its generalizability.
3. The paper provides extensive experiments across multiple real-world datasets (Pokec-n, Pokec-z, GitHub), showing that FairGuide achieves competitive performance in mitigating bias while maintaining utility.

## **Weaknesses**

**1. Motivation and Logic Flow**.
The logical flow of the Introduction is disjointed and the motivation for the specific design choices is not fully convincing:

- In Paragraph 3, the authors assert the importance of adding links. However, Paragraph 4 reviews pre/in/post-processing methods and merely states that structure updating is absent in in/post-processing. This does not sufficiently justify why a pre-processing approach is necessary or superior. The introduction lacks a critical analysis of the limitations of existing pre-processing methods, which makes the specific choice of "link addition" feel arbitrary.

**2. Impact of Artificial Links on Graph Properties**.
The method introduces "artificial" edges that may not represent ground-truth correlations. The paper lacks a discussion on the potential side effects of these modifications:

- **Structural Integrity:** How do these synthetic links affect the inherent properties of the graph, such as homophily or community structure?
- **Downstream Impact:** Is there a risk that these "false correlations" could mislead downstream GNNs, potentially degrading performance on specific tasks despite improved fairness metrics? A deeper analysis or ablation study on the quality of these added links is needed.

**3. Generalizability Beyond Binary Tasks**.
The current formulation and experiments focus on binary tasks. It is unclear whether the proposed meta-gradient framework and the fairness objective can be straightforwardly extended to multi-class classification or regression tasks. A discussion on the scalability of the method to more complex settings is missing.

**4. Inaccurate Characterization of Meta-Gradients**.
The description of the role of meta-gradients is technically imprecise and requires careful revision.

- The text implies that meta-gradients control the update process. In reality, they serve as a scoring function (a "referee") to estimate the potential gain of a candidate edge, but they do not inherently manage the updating execution.
- The statement in Section 4, *"To fully utilize the limited budget... FairGuide employs meta-gradients..."*, establishes a confusing causal link. Meta-gradients quantify the *effectiveness* of an edge, while the *limited budget* is a hard constraint handled by a selection strategy (e.g., Top-K). The gradient calculation itself is independent of the budget size.

**Additional Comments:**

Please see the above comments.

**Audience:**

Yes

**Audience Explanation:**

Adding links in the graph to mitigate bias and improve fairness is interesting. I think the fairness community would be interested to see the findings of this paper.

**Broader Impact Concerns:**

There are no concerns regarding the ethical implications of the work.

**Claims And Evidence:**

Yes

**Claims Explanation:**

This paper claims that adding new links to update graph structures is a good strategy to mitigate the fairness issue. Then, this paper proposes FairGuide, which introduces a community detection task as a pseudo task and calculates meta-gradients to assist the link addition process. Extensive experiments verify that the proposed framework indeed mitigates the bias issue of graphs. Therefore, the problem, the method, and the experiments are overall consistent.

**Requested Changes:**

There are some key changes that have to be made to meet the acceptance bar.

1. The visualization in Fig. 1 is somewhat abstract. Please provide a more detailed explanation of the specific bias pattern. Explicitly pointing out the bias in the figure would strengthen the motivation.
2. Please provide discussions or empirical results on how the added artificial links affect graph homophily. Do these links introduce noise that harms utility in certain regimes?
3. Please revise the claims regarding meta-gradients to accurately reflect their role.

---

> ### Author Response · Authors · 2026-07-12
>
> We thank the reviewer for recognizing our topological contribution and for the valuable suggestions.
>
> ### Response to Weakness 1: Motivation and Fair Guidance Problem
>
> We have substantially revised the Introduction to make the motivation clearer. The revised version now explains why existing model-level debiasing or heavy preprocessing does not directly address our goal of guiding the user graph itself. We also explicitly define the fair guidance problem and motivate link addition as a practical intervention for editable user graphs, since it can improve structural exposure while preserving observed links and node attributes. Please see the revised Introduction for the full motivation flow. We have also revised the Fig.1 to make the bias pattern of user graph clearer.
>
>
>
> ### Response to Weakness 2: Graph Structural Integrity and Link Prediction
>
> We added graph structural analyses to examine how FairGuide affects the graph data. We report sensitive homophily, label homophily, Louvain modularity, and largest-community ratio before and after FairGuide:
>
> | Dataset | Graph | Sensitive Homophily | Label Homophily | Modularity | Largest Community Ratio |
> |---|---|---:|---:|---:|---:|
> | GitHub | Original | 0.661 | 0.765 | 0.474 | 0.248 |
> | GitHub | FairGuide | 0.613 | 0.743 | 0.456 | 0.211 |
> | Pokec-n | Original | 0.953 | 0.591 | 0.567 | 0.271 |
> | Pokec-n | FairGuide | 0.870 | 0.587 | 0.547 | 0.242 |
> | Pokec-z | Original | 0.951 | 0.587 | 0.574 | 0.288 |
> | Pokec-z | FairGuide | 0.921 | 0.586 | 0.564 | 0.266 |
>
> FairGuide reduces sensitive homophily, which is consistent with improving fairness-relevant structural exposure. At the same time, label homophily changes only slightly, modularity decreases moderately, and the largest-community ratio does not increase. These results suggest that the added links improve fairness-relevant connectivity without collapsing the graph into an overly dominant community or substantially disrupting label-related connectivity.
>
> We also added link prediction on held-out original edges. The positive validation/test edges are sampled from the original graph; FairGuide-added links are used only as message-passing topology and are not treated as positive labels.
>
> | Dataset | Graph | AUC ↑ | AP ↑ |
> |---|---|---:|---:|
> | GitHub | Original | 0.865 $\pm$ 0.013 | 0.874 $\pm$ 0.014 |
> | GitHub | FairGuide | 0.852 $\pm$ 0.018 | 0.866 $\pm$ 0.018 |
> | Pokec-n | Original | 0.757 $\pm$ 0.033 | 0.722 $\pm$ 0.029 |
> | Pokec-n | FairGuide | 0.760 $\pm$ 0.032 | 0.758 $\pm$ 0.032 |
>
> The results show comparable link-prediction performance after applying FairGuide, supporting that the guided topology remains useful for structure-based prediction.
>
> To further connect this structural analysis with semantic plausibility, we conducted an LLM-based role-play experiment to evaluate whether the added edges are meaningful. The LLM judged 301 out of 500 FairGuide-added GitHub links as semantically meaningful.
>
> ### Response to Weakness 3: Generalization of Downstream Tasks
>
> We agree that the manuscript should clarify applicability beyond binary classification. At the method level, FairGuide's candidate-edge scoring is not tied to a binary downstream label. The pseudo task is a multi-community task, and the downstream objective can be replaced by task-appropriate differentiable objectives and fairness metrics.
>
> Empirically, we added both additional downstream-label classification experiments and a regression experiment. For classification, we keep the same sensitive attributes and replace the downstream label:
>
> | Dataset | New Target | Graph | $\Delta_{SP}$ (%) ↓ | $\Delta_{EO}$ (%) ↓ |
> |---|---|---|---:|---:|
> | Pokec-n | `I_like_music_indicator` | Original | 12.4 $\pm$ 2.7 | 7.7 $\pm$ 2.0 |
> | Pokec-n | `I_like_music_indicator` | FairGuide | **5.5 $\pm$ 3.3** | **2.7 $\pm$ 2.4** |
> | GitHub | `public_repos` | Original | 13.4 $\pm$ 1.6 | 17.7 $\pm$ 2.5 |
> | GitHub | `public_repos` | FairGuide | **10.9 $\pm$ 0.8** | **15.6 $\pm$ 1.4** |
>
> For regression task, we directly predict raw GitHub follower count and evaluate fair-regression metrics following prior fair-regression work (Agarwal et al., 2019; Chzhen et al., 2020; Gursoy and Kakadiaris, 2022):
>
> | Graph | Group MAE Gap ↓ | Prediction W1 ↓ | Prediction KS ↓ | Abs. Error W1 ↓ | Abs. Error KS ↓ |
> |---|---:|---:|---:|---:|---:|
> | Original | 45.4 $\pm$ 5.9 | 80.7 $\pm$ 5.5 | 0.16 $\pm$ 0.01 | 61.4 $\pm$ 5.6 | 0.13 $\pm$ 0.01 |
> | FairGuide | **5.2 $\pm$ 0.5** | **51.1 $\pm$ 0.6** | **0.10 $\pm$ 0.01** | **14.7 $\pm$ 0.5** | **0.02 $\pm$ 0.01** |
>
> These experiments strengthen the evidence that FairGuide can improve fairness beyond the original binary classification setting.
>
> ### Response to Weakness 4: Meta-Gradient Wording
>
> We revised the methodology to clarify that meta-gradients are used only as candidate-link scoring signals. The discrete and budgeted graph update is performed by the subsequent Gumbel-based top-$k$ selection step, not by the gradient itself.